# The kernel of graph indices for vector search

## Abstract

The most popular graph indices for vector search use principles from computational geometry to build the graph. Hence, their formal graph navigability guarantees are only valid in Euclidean space. In this work, we show that machine learning can be used to build graph indices for vector search in metric and non-metric vector spaces (e.g., for inner product similarity). From this novel perspective, we introduce the Support Vector Graph (SVG), a new type of graph index that leverages kernel methods to establish the graph connectivity and that comes with formal navigability guarantees valid in metric and non-metric vector spaces. In addition, we interpret the most popular graph indices, including HNSW and DiskANN, as particular specializations of SVG and show that new indices can be derived from the principles behind this specialization. Finally, we propose SVG-L0 that incorporates an $\ell_0$ sparsity constraint into the SVG kernel method to build graphs with a bounded out-degree. This yields a principled way of implementing this practical requirement, in contrast to the traditional heuristic of simply truncating the out edges of each node. Additionally, we show that SVG-L0 has a self-tuning property that avoids the heuristic of using a set of candidates to find the out-edges of each node and that keeps its computational complexity in check.

## 1 Introduction

In recent years, vector search has become a critical component of AI infrastructure. For example, in retriever-augmented generation (RAG) (Lewis et al., 2020), vector search is used to ground knowledge and prevent hallucinations. In particular, graph-based indices (e.g., Dearholt et al., 1988; Arya & Mount, 1993; Malkov & Yashunin, 2020; Fu et al., 2019; Subramanya et al., 2019) have been deployed in the real world with great success. Here, a directed graph, where each vertex corresponds to a database vector and edges represent neighbor-relationships between vectors, is efficiently traversed to find the (approximate) nearest neighbors of a query vector in sublinear time. The graph edges need to be carefully selected to ensure that this traversal yields correct results (i.e., the graph is navigable). Starting with the seminal works by Dearholt et al. (1988) and Arya & Mount (1993), navigable graphs are built using computational geometry principles to perform edge selection. The Delaunay graph is a fully navigable triangulation (Wang et al., 2021) but it is too dense in higher dimensions. Informally, graph-building algorithms sparsify the graph by examining these triangles and only keeping a small subset. The Delaunay graph and the triangle pruning rules are defined in Euclidean space, which limits the navigability guarantees of the resulting graphs to this specific case. However, these algorithms are commonly used in non-Euclidean spaces, where their underlying principles do not hold, to build graphs that work well in practice but lack formal guarantees.

In this work, we analyze graph indices from a new perspective: we rely on machine learning instead of computational geometry. In particular, we study graph indices formally in light of kernel methods. In this setting, we have a similarity function $\mathrm{sim}(\mathbf{x}, \mathbf{x}') : \mathbb{R}^d \times \mathbb{R}^d \to \mathbb{R}$ and an associated kernel $K(\mathbf{x}, \mathbf{x}') = h(\mathrm{sim}(\mathbf{x}, \mathbf{x}'))$, where $h$ is a (possibly) nonlinear function. Throughout this work, we assume that the kernel is positive semidefinite, i.e., the feature expansion $K(\mathbf{x}, \mathbf{x}') = \phi(\mathbf{x})^\top \phi(\mathbf{x}')$ is valid for (possibly infinite-dimensional) feature vectors $\phi(\mathbf{x})$. The exponential kernel is an important class of kernels, widely used in experimental and theoretical studies,

$$K_{\mathrm{EXP}}\left(\mathbf{x}, \mathbf{x}'\right) \stackrel{\mathrm{def}}{=} \exp\left(\mathrm{sim}(\mathbf{x}, \mathbf{x}')/\sigma^2\right), \tag{1}$$

where the hyperparameter $\sigma > 0$ is implicit. Two leading examples are given by the similarity functions

$$\text{sim}_{\text{EUC}}(\mathbf{x}, \mathbf{x}') \stackrel{\text{def}}{=} - \left\| \mathbf{x} - \mathbf{x}' \right\|_2^2 , \tag{2}$$

$$\text{sim}_{\text{DP}}(\mathbf{x}, \mathbf{x}') \stackrel{\text{def}}{=} \mathbf{x}^\top \mathbf{x}' \tag{3}$$

that define the standard Radial Basis Function (a.k.a. Gaussian) and exponential dot product kernels, respectively. The similarity $\text{sim}_{\text{DP}}$ corresponds to the commonly used maximum inner product (MIP) retrieval problem. The exponential kernel is a natural choice, as it is used in the training loss (e.g., the entropy loss) of the embedding models that produce the vectors commonly used in practice (Radford et al., 2021; Karpukhin et al., 2020). Note that defining $\text{sim}(\mathbf{x}, \mathbf{x}') \stackrel{\text{def}}{=} - \text{dist}(\mathbf{x}, \mathbf{x}')^2$ for any distance function (e.g., Manhattan, Hamming, etc.) yields a valid exponential kernel.

Using kernels as our vantage point, we present the following contributions (all proofs in the appendix):

- We propose a new graph index, the Support Vector Graph (SVG), with formal navigability guarantees (under some conditions). This new index arises from a novel perspective on graph indices (Section 2) and an accompanying formulation that models graph construction as a kernelized nonnegative least squares (NNLS) problem. This NNLS is equivalent to a support vector machine (SVM) whose support vectors provide the connectivity of the graph (Section 3).

- We introduce the concept of quasi-navigable networks (Section 3.1). Navigable networks are designed for the exact nearest neighbor (NN) problem in Euclidean space but are commonly used to find approximate nearest neighbors (ANN) in non-Euclidean spaces. Quasi-navigable networks rely on kernels, thus covering many metric and non-metric spaces, and offer a relaxed definition of navigability more attuned to ANN.

- For SVG, our formal results, valid in metric and non-metric vector spaces, show quasi-navigability for general kernels and navigability for the exponential kernel with small width $\sigma$ (Section 3.1). To the best of our knowledge, these are the first navigability results in non-Euclidean vector spaces.

- We formally interpret the most popular graph indices as SVG specializations, where the SVG optimization problem is used within the aforementioned traditional triangle pruning approach (Section 4). In particular, our results cover the popular HNSW (Malkov & Yashunin, 2020) and DiskANN (Subramanya et al., 2019). We also show that new triangle-pruning algorithms, valid in Euclidean and non-Euclidean spaces, can be derived from the principles behind this specialization.

- Finally, we address the construction of graphs with a bounded out-degree, a common feature in most practical deployments. For this, we propose SVG-L0 that includes a hard sparsity ($\ell_0$) constraint into the SVG optimization (Section 5). SVG-L0 yields a principled way of handling the requirement, in contrast with the traditional heuristic, which simply truncates the out edges of each node. Additionally, we show that SVG-L0 has a self-tuning property, which avoids setting a set of candidate edges for each graph node and makes its computational complexity sublinear in the number of indexed vectors.

Although this work focuses on a formal analysis of SVG and SVG-L0, we also present some preliminary empirical results to show that the proposed techniques have practical value beyond their theoretical merits (Section 6). For reproducibility, we make our implementation available at `[anonymized_url]`.

**Notation.** We denote the set of natural numbers from 1 to $n$ by $[1 \ldots n]$. We denote vectors/matrices by lowercase/uppercase bold letters, e.g., $\mathbf{v} \in \mathbb{R}^n$ and $\mathbf{A} \in \mathbb{R}^{m \times n}$. Individual entries of a matrix $\mathbf{A}$ (resp. vector $\mathbf{v}$) are denoted by $\mathbf{A}_{[ij]}$ (resp. $v_i$). The $i$-th row and column of $\mathbf{A}$ are denoted by $\mathbf{A}_{[i:]}$ and $\mathbf{A}_{[:i]}$, respectively. The matrix containing a subset $\mathcal{I} \subset [1 \ldots m]$ (resp. $\mathcal{J} \subset [1 \ldots n]$) of the rows (resp. columns) of $\mathbf{A} \in \mathbb{R}^{m \times n}$ is denoted by $\mathbf{A}_{[\mathcal{I}:]}$ (resp. $\mathbf{A}_{[:\mathcal{J}]}$). A directed graph $G = ([1 \ldots n], \mathcal{E})$ is composed by the node set $[1 \ldots n]$ and the edge/vertex set $\mathcal{E}$, i.e., a set of ordered pairs $\overrightarrow{ij}$ with $i, j \in [1 \ldots n]$. We define the neighborhood of node $i$ as $\mathcal{N}_i \stackrel{\text{def}}{=} \left\{ j \mid \overrightarrow{ij} \in \mathcal{E} \right\}$. A path $[v_1, \cdots, v_l]$ in $G$ is a list of nodes such that $(\forall i = 1, \cdots, l-1) \; \overrightarrow{v_i v_{i+1}} \in \mathcal{E}$.

## 2 Graph indices for vector search in Euclidean Space

Using navigable graphs for vector search has a long history (Dearholt et al., 1988; Arya & Mount, 1993) but only became prominent in the last ten years (e.g., Subramanya et al., 2019; Malkov & Yashunin, 2020)

---

**Algorithm 1:** Greedy graph search

    **Input**   : Query $\mathbf{q} \in \mathbb{R}^d$, dataset $\left\{\mathbf{x}_i \in \mathbb{R}^d\right\}_{i=1}^n$, graph $G = ([1 \dots n], \mathcal{E})$, entry point $i_{\text{ep}}$.

    **Output:** Approximate nearest neighbor $i^*$.

**1** $i^* \leftarrow i_{\text{ep}}$;

**2 Repeat**

**3**     $i \leftarrow \underset{j \in \mathcal{N}_{i^*}}{\operatorname{argmin}} \|\mathbf{q} - \mathbf{x}_j\|_2$; // $\mathcal{N}_{i^*}$ is the neighborhood of $i^*$

**4**     **if** $\|\mathbf{q} - \mathbf{x}_i\|_2 < \|\mathbf{q} - \mathbf{x}_{i^*}\|_2$ **then** $i^* \leftarrow i$; // progress, continue

**5**     **else return** $i^*$; // no progress, exit

---

with the increasing scale of unstructured data. Navigability is defined as the ability to reach any node when conducting a greedy graph traversal (Algorithm 1) using that node as the query. The following definitions formalize this concept.

**Definition 1** (Monotonic Path (Fu et al., 2019))**.** *Given a set of n vectors $\left\{\mathbf{x}_i \in \mathbb{R}^d\right\}_{i=1}^n$, let $G = ([1 \dots n], \mathcal{E})$ denote a directed graph and $s, t \in [1 \dots n]$ be two nodes of $G$. A path $[v_1, \cdots, v_l]$ from $s = v_1$ to $t = v_l$ in $G$ is a monotonic path if and only if $(\forall i = 1, \cdots, l-1) \|\mathbf{x}_{v_i} - \mathbf{x}_t\|_2 > \|\mathbf{x}_{v_{i+1}} - \mathbf{x}_t\|_2$.*

**Definition 2** (Monotonic Search Network (Fu et al., 2019))**.** *Given a set of n vectors $\left\{\mathbf{x}_i \in \mathbb{R}^d\right\}_{i=1}^n$ and the kernel $K$, a graph $G = ([1 \dots n], \mathcal{E})$ is a generalized monotonic search network if and only if there exists at least one monotonic path from s to t for any two nodes $s, t \in [1 \dots n]$.*

A Monotonic Search Network (MSNet) is a navigable graph as demonstrated by the following lemma.

**Lemma 1** (Fu et al., 2019)**.** *Let $G = ([1 \dots n], \mathcal{E})$ be a monotonic search network. Let $s, t \in [1 \dots n]$, then Algorithm 1 with $\mathbf{x}_t$ as the query and s as the entry point finds a monotonic path from s to t in $G$.*

The Delaunay graph (DG) is an MSNet (Kurup, 1992). For a graph with $n$ nodes, the number of edges in the DG rapidly approaches $n$ as the dimensionality grows, limiting its usability for large datasets with high-dimensional vectors (the memory and computational complexities approach $O(n^2)$ and $O(n)$, respectively). As a consequence, many graph construction algorithms (e.g., Dearholt et al., 1988; Arya & Mount, 1993; Malkov & Yashunin, 2020; Fu et al., 2019; 2022; Subramanya et al., 2019) have been proposed over the years, operating under the (sometimes implicit) principle of sparsifying the DG. These algorithms work as depicted in Algorithm 2. For each node $i$, a candidate pool is selected (this is commonly implemented as an approximate nearest neighbor search), and then a pruning algorithm is used to select a maximally diverse set of nodes (i.e., far away from each other) while being close to $i$, see Figure 1. In essence, these algorithms were carefully conceived to analyze the DG triangles (or a superset (Subramanya et al., 2019)) and discard those edges that are redundant for navigability. While these graphs were designed to have formal navigability guarantees when $\mathcal{C}_i = [1 \dots n] \setminus \{i\}$, they are lost when $\mathcal{C}_i \subset [1 \dots n] \setminus \{i\}$.

More importantly, the DG and the main graph indices (e.g., Malkov & Yashunin, 2020; Fu et al., 2019; Subramanya et al., 2019) rely on principles from computational geometry, such as triangular inequalities and (as we show in Section 4) on the law of cosines, which are only valid in Euclidean space. These graph indices have been used in non-Euclidean vector spaces by extending their edge pruning rules to other similarities in an ad hoc fashion, resulting in a lack of understanding of their practical behavior.

### 2.1 From graph search to multiclass classification

We now adopt an alternative viewpoint that will ultimately lead to new developments. For this, we think of a greedy search in DG, the original monotonic search network, using Algorithm 1 as finding an ascending path in a multiclass classification problem.

The Voronoi diagram is a tessellation of the space, where each node $i$ of the DG corresponds to a distinct convex cell $C_i$ (see Figure 14 in the appendix). Two nodes are connected in the DG if the corresponding Voronoi cells share a facet. We associate with each cell $C_i$ a decision function $f_i : \mathbb{R}^d \to \mathbb{R}$ such that: $\mathbf{x} \in C_i$ if $f_i(\mathbf{x}) \geq 0$, $\mathbf{x} \notin C_i$ if $f_i(\mathbf{x}) < 0$, and $f_i(\mathbf{x})$ decreases as the distance between $\mathbf{x}$ and $C_i$, $\min_{\mathbf{x}' \in C_i} \|\mathbf{x} - \mathbf{x}'\|_2$,

**Algorithm 2:** Graph index construction

> **Input** : Dataset $\left\{\mathbf{x}_i \in \mathbb{R}^d\right\}_{i=1}^n$.
> **Output:** Graph $G = ([1\ldots n], \mathcal{E})$.

**1** $\mathcal{E} \leftarrow \emptyset$;
**2 for** $i \in [1\ldots n]$ **do**
**3**      **Selection:** choose a candidate pool $\mathcal{C}_i \subseteq [1\ldots n] \setminus \{i\}$;
**4**      **Pruning:** create set $\mathcal{N}_i \subseteq \mathcal{C}_i$ containing the out neighbors of node $i$ by applying a pruning algorithm;
**5**      $\mathcal{E} \leftarrow \mathcal{E} \cup \left\{\vec{ij} \mid j \in \mathcal{N}_i\right\}$;

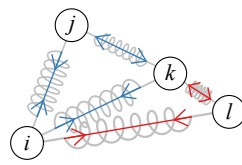

Figure 1: Conceptual depiction of the pruning strategy in Euclidean space to find the out-edges of node $i$ in the graph index $G = ([1\ldots n], \mathcal{E})$. Attractive (inward arrowheads) and repulsive (outward arrowheads) forces promote similarity with $i$ or diversity between candidates, respectively. Blue and red arrows depict favorable and less favorable forces, respectively. Here, $\{\vec{ij}, \vec{ik}\} \subset \mathcal{E}$ but $\vec{il} \notin \mathcal{E}$ as one can move from $i$ to $k$ and then from $k$ to $l$ using the greedy search in Algorithm 1.

increases. Each $f_i$ is determined by the intersection of the half-spaces of the boundaries of $C_i$. Finding the nearest neighbor of a query $\mathbf{q}$ is equivalent to finding $i$ such that $f_i(\mathbf{q}) \geq 0$. This corresponds to a multiclass classification problem with $n = |\mathcal{X}|$.[1] Of course, this is not computationally very useful, as we are evaluating $n$ classifiers (i.e., scanning the entire set $\mathcal{X}$). Seeking fewer evaluations, we conceptualize Algorithm 1 as follows: if $f_{i*}(\mathbf{q}) \geq 0$, the vector $\mathbf{x}_{i*}$ is the nearest neighbor of $\mathbf{q}$; if $f_{i*}(\mathbf{q}) < 0$, move to the adjacent cell $i$ so that $f_i(\mathbf{q})$ is maximum. That is, instead of directly solving the multiclass classification problem, we use the decision functions of adjacent cells (i.e., given by the Delaunay edges) to find an ascending path $[v_1, \cdots, v_l]$ such that $f_{v_i} < f_{v_{i+1}}$. Through this ascent algorithm, we only evaluate a small subset of the $n$ classifiers.

This qualitative viewpoint raises several questions. Can we use machine learning (ML) to build graph indices? And in non-Euclidean spaces? Can we leverage ML to build parsimonious graphs? Is there a connection between the ML approach and "traditional" graph indices? In the remainder of this paper, we answer these questions in the affirmative, using other classifiers, i.e., support vector machines, to develop new graph indices in Euclidean and non-Euclidean spaces, with the properties and contributions discussed in the introduction.

## 3 The Support Vector Graph

We now define a new type of graph inspired by the ideas outlined in the previous section. Instead of relying on principles from computational geometry, we directly leverage the result of an optimization algorithm, the nonnegative least squares problem in kernel space. The positive semidefinite kernel matrix $\mathbf{K}$ with entries $\mathbf{K}_{[ij]} = K(\mathbf{x}_i, \mathbf{x}_j) = \phi(\mathbf{x}_i)^\top \phi(\mathbf{x}_j)$ can be written as $\mathbf{K} = \mathbf{\Phi}^\top \mathbf{\Phi}$ where $\mathbf{\Phi} = [\phi(\mathbf{x}_1), \cdots, \phi(\mathbf{x}_n)]$, with the vectors horizontally stacked. From now on, and unless otherwise specified, the similarity between two vectors $\mathbf{x}_i$ and $\mathbf{x}_j$ will only be determined by the value of $K(\mathbf{x}_i, \mathbf{x}_j)$. With these elements, we present the proposed graph index.

**Definition 3.** *We define the Support Vector Graph (SVG) as the result of connecting node $i$ to the non-zero elements of the minimizer $\mathbf{s}^{(i)}$ of*

$$\min_{\mathbf{s}} \frac{1}{2} \left\| \phi(\mathbf{x}_i) - \mathbf{\Phi}\mathbf{s} \right\|_2^2 \quad s.t. \quad \mathbf{s} \geq \mathbf{0}, \; s_i = 0, \tag{4}$$

*where $\mathbf{\Phi} = [\phi(\mathbf{x}_1), \cdots, \phi(\mathbf{x}_n)]$.*

When $K(\mathbf{x}_i, \mathbf{x}_j) = \phi(\mathbf{x}_i)^\top \phi(\mathbf{x}_j)' \geq 0$ for any $i, j$ (a common choice), the angle between any pair of vectors is less than $\pi/2$ and we can find a rotation such that all the feature vectors lie in the positive orthant. Then, the columns of $\mathbf{\Phi}$ satisfy the conditions that ensure a unique solution (e.g., Wang & Tang, 2009; Slawski & Hein, 2011; 2014) (i.e., preventing the nonnegative least squares from overfitting). Moreover, in this setting,

---

[1]Inverted indices use a similar computational motif derived from Voronoi diagrams (Jégou et al., 2011).

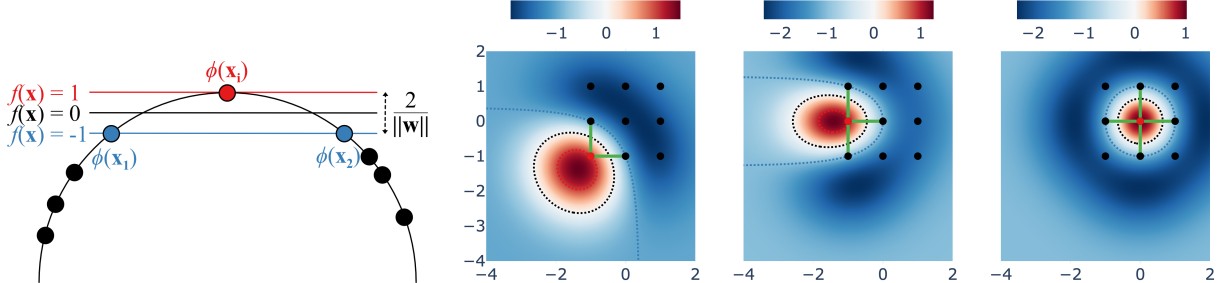

Figure 2: (Left) Conceptual representation of the SVM hyperplane and margins involved in SVG. Here, the vector $\mathbf{x}_i$ is connected to its support vectors $\mathbf{x}_1$ and $\mathbf{x}_2$ for which $f_i(\mathbf{x}_1) = f_i(\mathbf{x}_2) = -1$, see Equation (7). (Right) Example of the SVM decision function values (the level sets $f_i(\mathbf{x}) = 1, 0, -1$ are marked in dotted red, black and blue lines, respectively). We observe that the function $f_i$ adjusts its shape to the topology of its surrounding points (i.e., the area where $f_i(\mathbf{x}) > 0$ adapts to its surroundings).

Problem (4) is self-regularizing, in the sense that its minimizer is naturally sparse (Slawski & Hein, 2011) without including any explicit constraints promoting sparsity (in contrast to the DG whose sparsity depends on the input dimensionality, i.e., less sparse at higher dimensions).

The use of sparsity-regularized regression problems to build graphs is not new in machine learning. Some notable applications include the estimation of sparse inverse covariance matrices (Meinshausen & Bühlmann, 2006), subspace learning and clustering (Cheng et al., 2010; Hosseini & Hammer, 2018), spectral clustering (Xiao et al., 2012), and nonnegative matrix factorization (for bipartite graphs) (Kumar et al., 2013). In particular, a variant of Problem (4), which relies on the selection of a candidate pool as in Algorithm 2, was used for manifold learning (Shekkizhar & Ortega, 2023). However, our analysis of graphs built with nonnegative sparse regression for vector search is new.

Furthermore, SVG establishes an interesting link between graph indices and parsimonious vector coding. That is, with a linear kernel, the loss in Problem (4) becomes $\frac{1}{2} \|\mathbf{x}_i - \mathbf{X}\mathbf{s}\|_2^2$, where $\mathbf{X} = [\mathbf{x}_1, \cdots, \mathbf{x}_n]$. This formulation is commonly used to represent (e.g., Elhamifar et al., 2012) and quantize vectors (in additive quantization (Martinez et al., 2016), for example). There is a conceptual parallelism with inverted indices (Jégou et al., 2011), which are derived from vector quantizers (k-means).

As we show next, the connection between Problem (4) and navigable graphs starts emerging as we dig deeper into the problem's properties. By analyzing the expanded form of Problem (4),

$$\min_{\{s_j\}_{j=1}^n} \frac{1}{2} K(\mathbf{x}_i, \mathbf{x}_i) + \underbrace{\frac{1}{2} \sum_{j,k \neq i} s_j s_k K(\mathbf{x}_j, \mathbf{x}_k)}_{\text{term A}} - \underbrace{\sum_{j \neq i} s_j K(\mathbf{x}_i, \mathbf{x}_j)}_{\text{term B}} \quad \text{s.t.} \quad (\forall j)\, s_j \geq 0,\; s_i = 0,\; \sum_{j \neq i} s_j = 1, \quad (5)$$

it becomes clear that its solution balances diversity (repulsion) and similarity (attraction) forces using similar principles as those shown in Figure 1 and analyzed in detail in Section 4 for other popular graph indices. The minimization of term A promotes the selection of a diverse set of edges, i.e., indices $j, k$ such that $K(\mathbf{x}_j, \mathbf{x}_k)$ is small. When using the RBF kernel, it favors out-neighbors that are far away from each other. The minimization of term B promotes the selection of edges that are similar to $\mathbf{x}_i$, i.e., indices $j$ such that $K(\mathbf{x}_i, \mathbf{x}_j)$ is high. When using the RBF kernel, it favors out-neighbors that are close to $\mathbf{x}_i$.

In the previous section, we qualitatively connected graph indices with a multi-class classification problem. It turns out that Problem (4) is a classification problem disguised as a regression problem.

**Theorem 1.** *Problem (4) is equivalent to a hard-margin support vector machine classifier using the labels*

$$y_j = \begin{cases} 1 & \text{if } j = i, \\ -1 & \text{otherwise.} \end{cases} \quad (6)$$

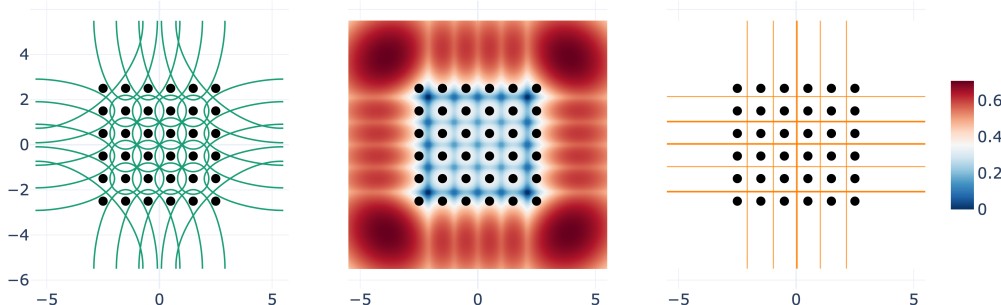

Figure 3: The SVM decision boundaries (left), i.e., $f_i(\mathbf{x}) = 0$, for each point in a regular 2D grid; see Equation (7). The function $f(\mathbf{x}) = \max_i f_i(\mathbf{x})$ (center) induces a tessellation. As expected, the tessellation, found by running a watershed algorithm on $f(\mathbf{x})$, forms a regular grid (right).

*The nonzero elements of the minimizer $\mathbf{s}^{(i)}$ of Problem (4) are the support vectors.*

We refer the reader to Appendix A for a quick primer on SVMs. For the $i$-th vector, after computing the minimizer $\mathbf{s}^*$ to Problem (4) we obtain the SVM decision function

$$f_i(\mathbf{x}) = \frac{\mathbf{w}_i^\top \phi(\mathbf{x}) + b_i}{\mathbf{w}_i^\top \phi(\mathbf{x}_i) + b_i} \quad \text{where} \quad \mathbf{w}_i = \phi(\mathbf{x}_i) - \sum_{j \neq i} s_j^* \phi(\mathbf{x}_j), \quad b_i = -\frac{1}{2}\mathbf{w}_i^\top \left( \phi(\mathbf{x}_i) + \phi(\mathbf{x}_{j'}) \right), \qquad (7)$$

for any $j'$ such that $s_{j'}^* > 0$. By construction, $f_i(\mathbf{x}_i) = 1$ and $f_i(\mathbf{x}_j) = -1$ for every $j \in \mathcal{N}_i$, and $f_i(\mathbf{x}_j) < -1$ for every $j \in [1..n] \setminus \mathcal{N}_i$. In Figure 2 (left), we present a conceptual representation of these level sets as hyperplanes in feature space. Figure 2 (right) illustrates that these level sets materialize in the original space as nonlinear boundaries that adapt their shape to the topology of the vectors surrounding $\mathbf{x}_i$.

This alternative formulation of Problem (4) makes it easy to see why $|\mathcal{N}_i| \ll n$: The support vector set $\mathcal{N}_i$ is sparse in separable and non-degenerate settings. More importantly, the decision functions $f_i$ induce a tessellation of the space, as observed in Figure 3. We can find such a tessellation by considering the function $F(\mathbf{x}) = -\max_i f_i(\mathbf{x})$ as a topographic map and separating adjacent catchment basins (following its gradient) using a watershed algorithm (Couprie & Bertrand, 1997). The link between the SVG and this tessellation is analogous to that of the Delaunay graph and the Voronoi diagram.

SVG also shares a deep connection with the DG. Other graph indices are subgraphs of the DG by applying pruning rules to its edges. As shown next, when using a kernel based on the Euclidean distance (e.g., the RBF kernel), SVG sparsifies the DG by solving optimization problems (see Figures 4 and 5).

**Theorem 2.** *Let $G$ be the Delaunay graph computed from the original vectors $\{\mathbf{x}_i\}_{i=1}^n$. When using a kernel based on the Euclidean distance (e.g., RBF), the support of the solution to Problem (4) is a subset of the neighbors of node $i$ in $G$.*

### 3.1 Navigability

So far, we have described how to build an SVG and how it shares some key properties with the DG and other graph indices. We now turn our attention to the analysis of its navigability. However, before proceeding, we provide new definitions of graph navigation that conform to the kernelized setup of the problem.

In our setup, the Euclidean Algorithm 1 is transformed into Algorithm 3 by switching the minimization of the distance with the maximization of the kernel values. In the following, we provide definitions that are analogous to Definitions 1 and 2 but are adapted to the use of kernels. It is important to note that, with non-metric similarities, the terminal node may be different from the query. That is, whereas we always have $\text{sim}_{\text{EUC}}(\mathbf{x}_k, \mathbf{x}_k) = \max_{i=1,\dots n} \text{sim}_{\text{EUC}}(\mathbf{x}_i, \mathbf{x}_k)$, it is possible that $\text{sim}_{\text{DP}}(\mathbf{x}_k, \mathbf{x}_k) < \max_{i=1,\dots n} \text{sim}_{\text{DP}}(\mathbf{x}_i, \mathbf{x}_k)$ (see Equations (2) and (3)).

Navigable graphs enable retrieving the 1-NN exactly. However, these graphs are commonly used for approximate retrieval and, although the strict definition of navigability may be compromised when building these

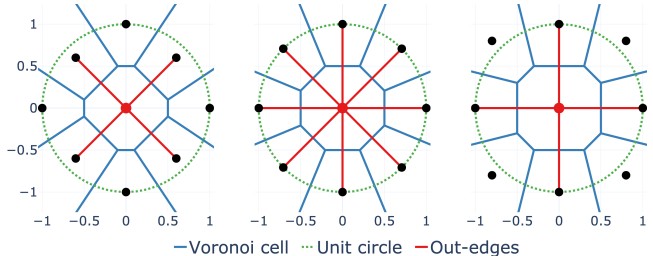

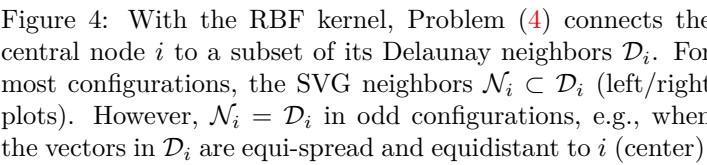

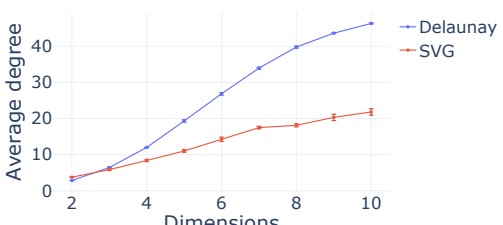

Figure 4: With the RBF kernel, Problem (4) connects the central node $i$ to a subset of its Delaunay neighbors $\mathcal{D}_i$. For most configurations, the SVG neighbors $\mathcal{N}_i \subset \mathcal{D}_i$ (left/right plots). However, $\mathcal{N}_i = \mathcal{D}_i$ in odd configurations, e.g., when the vectors in $\mathcal{D}_i$ are equi-spread and equidistant to $i$ (center).

Figure 5: The average cardinality of the SVG neighbors grows slower than that of the Delaunay neighbors for (ten realizations of) 100 randomly distributed vectors as their dimension grows.

---

**Algorithm 3:** Greedy graph search in kernel space

> **Input** : Query $\mathbf{q} \in \mathbb{R}^d$, dataset $\left\{\mathbf{x}_i \in \mathbb{R}^d\right\}_{i=1}^n$, graph $G = ([1\ldots n], \mathcal{E})$, entry point $i_{\mathrm{ep}}$.
> **Output:** Approximate nearest neighbor $i^*$.

**1** $i^* \leftarrow i_{\mathrm{ep}}$;
**2 Repeat**
**3** $\quad$ $i \leftarrow \underset{j \in \mathcal{N}_{i^*}}{\operatorname{argmax}} K(\mathbf{x}_j, \mathbf{q})$; // $\mathcal{N}_{i^*}$ is the neighborhood of $i^*$
**4** $\quad$ **if** $K(\mathbf{x}_i, \mathbf{q}) > K(\mathbf{x}_{i^*}, \mathbf{q})$ **then** $i^* \leftarrow i$; // progress, continue
**5** $\quad$ **else return** $i^*$; // no progress, exit

---

graph indices in practice (e.g., the fast preprocessing in Indyk & Xu, 2023), they still perform very well. In this sense, a relaxed definition of navigability may help fill the gap between the theoretical goals and the practical performance of graph indices. As such, we define the concept of quasi-navigable networks allowing for slack in the monotonicity of the paths. Later on, we provide empirical results showing that these networks can offer better retrieval than heuristically modified navigable networks. Note that this slack is analogous to the one used to formally define the ANN problem (e.g., Andoni & Indyk, 2008), where $(1 + \epsilon)$-NNs are considered for some $\epsilon > 0$. Understanding the guarantees offered by quasi-navigable graphs in the ANN setting is an interesting problem for future research.

**Definition 4** (Generalized $\epsilon$-Monotonic Path). *Given $\epsilon > 0$, a set of $n$ vectors $\left\{\mathbf{x}_i \in \mathbb{R}^d\right\}_{i=1}^n$ and the kernel $K$, let $G = ([1\ldots n], \mathcal{E})$ denote a directed graph, $s, k \in [1\ldots n]$ be two nodes of $G$, and $t = \underset{i=1,\ldots n}{\operatorname{argmax}} K(\mathbf{x}_i, \mathbf{x}_k)$. A path $[v_1, \cdots, v_l]$ from $s = v_1$ to $t = v_l$ in $G$ is a generalized $\epsilon$-monotonic path if and only if $(\forall i = 1, \cdots, l-1)$ $K(\mathbf{x}_{v_i}, \mathbf{x}_t) < (1+\epsilon) K(\mathbf{x}_{v_{i+1}}, \mathbf{x}_t)$.*

**Definition 5** (Generalized $\epsilon$-Monotonic Search Network). *Given a set of $n$ vectors $\left\{\mathbf{x}_i \in \mathbb{R}^d\right\}_{i=1}^n$ and the kernel $K$, a graph $G = ([1\ldots n], \mathcal{E})$ is a generalized $\epsilon$-monotonic search network if and only if there exists at least one generalized $\epsilon$-monotonic path from $s$ to $t = \underset{i=1,\cdots,n}{\operatorname{argmax}} K(\mathbf{x}_i, \mathbf{x}_k)$ for any two nodes $s, k \in [1\ldots n]$.*

When using a kernel based on the Euclidean distance, $k = \operatorname{argmax}_{i=1,\ldots n} K(\mathbf{x}_i, \mathbf{x}_k)$. In this setting, taking $\epsilon = 0$ recovers Definitions 1 and 2. To finalize the setup, we show that generalized 0-monotonic search networks are navigable using Algorithm 3.

**Lemma 2.** *Let $G = ([1\ldots n], \mathcal{E})$ be a generalized 0-monotonic search network. Let $s, k \in [1\ldots n]$, then Algorithm 3 with $\mathbf{x}_k$ as the query and $s$ as the entry point finds a generalized monotonic path from $s$ to $t = \underset{i=1,\cdots,n}{\operatorname{argmax}} K(\mathbf{x}_i, \mathbf{x}_k)$ in $G$.*

We now show that SVGs are quasi-navigable for general kernels and navigable for exponential kernels with a small width. As our kernel-based definitions subsume arbitrary distance functions and even non-metric

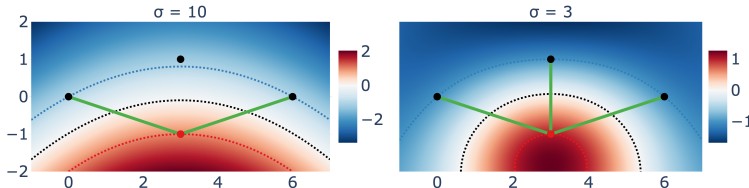

Figure 6: The width of the RBF kernel acts as a regularizer of the SVG connectivity (i.e., it smooths the SVM boundaries). For $\sigma = 10$, the top center point is not included among the neighbors of the red point, while for $\sigma = 3$, it is. We observe that the curvature of the boundary grows as $\sigma$ decreases.

similarities (like the inner product), this is the first theoretical result for graph navigability in non-Euclidean spaces.

**Theorem 3.** *A SVG is a generalized $\epsilon$-monotonic search network with*

$$\epsilon = \max_{i \in [1..n]} \epsilon_i \quad where \quad \epsilon_i = \max\left\{\mathbf{1}^\top \mathbf{s}^{(i)}, 1\right\} - 1, \tag{8}$$

*where $\mathbf{s}^{(i)}$ is the minimizer of Problem ([4]) for node i.*

**Theorem 4.** *When using an exponential kernel with width $\sigma^2$, the SVG is a generalized $\epsilon$-monotonic search network with*

$$\epsilon = \max_{i \in [1..n]} \epsilon_i \quad s.t. \quad \epsilon_i = \max\left\{\exp\left(\log\left\|\mathbf{s}^{(i)}\right\|_0 + \log s_j^{(i)}\right), 1\right\} - 1, \tag{9}$$

*where $\mathbf{s}^{(i)}$ is the minimizer of Problem ([4]) for node i.*

It is well known in kernel methods that the kernel width acts as a regularizer, i.e., a larger width smooths out the class boundary. It is only natural that selecting a large width may have such an effect on the SVG connectivity, which may create issues for its navigability. An example is observed in Figure 6. However, selecting a suitably small width fixes these issues, as shown next for exponential kernels.

**Corollary 1.** *When using an exponential kernel with width $\sigma^2$, the SVG is a generalized $0$-monotonic search network for $\sigma \to 0$.*

The following corollary follows immediately from Lemma 2 and Corollary 1.

**Corollary 2.** *When using an exponential kernel with width $\sigma^2$, the SVG is navigable using Algorithm 3 for $\sigma \to 0$.*

These results show that SVG, derived using completely different tools (i.e., kernel methods from machine learning) than existing graph indices, is a suitable graph index for vector search.

## 4 Connecting SVG to other graph indices

We now study the link between SVG and other popular graph indices. Algorithm 4 provides a blueprint for most pruning techniques (Malkov & Yashunin, 2020; Subramanya et al., 2019; Fu et al., 2022) used in Algorithm 2. Given a candidate pool $\mathcal{C}_i$, Algorithm 4 considers triplets of nodes $i, j, k$, as depicted in Figure 7. Throughout this section, we use the standard assumption $\mathcal{C}_i = [1 \dots n] \setminus \{i\}$ (we discuss this choice in the next section).

Next, we show that Problem ([4]), when applied to the analysis of these triplets, leads to a traditional graph sparsification algorithm with navigability guarantees for general kernels.

**Lemma 3** (Kernel connectivity rule). *In Line 6 of Algorithm 4, the connectivity rule derived from Problem ([4]) is: keep k in $\mathcal{C}_i$ if*

$$K(\mathbf{x}_i, \mathbf{x}_j)K(\mathbf{x}_j, \mathbf{x}_k) < K(\mathbf{x}_i, \mathbf{x}_k). \tag{10}$$

---

**Algorithm 4:** Pruning meta-algorithm to determine the outgoing edges for node $i$

---

**Input**  : Dataset $\mathcal{X} = \left\{\mathbf{x}_j \in \mathbb{R}^d\right\}_{j=1}^n$, node $i \in [1 \dots n]$, candidate pool $\mathcal{C}_i$, maximum out-degree $M \in \mathbb{N}^+$.

**Output:** Set $\mathcal{N}_i$ of outgoing neighbors for node $i$

**1** $\mathcal{E}_i \leftarrow \emptyset$;

**2** **while** $\mathcal{C}_i \neq \emptyset$ **do**

**3** $\quad j \leftarrow \underset{j' \in \mathcal{C}}{\operatorname{argmax}}\, K(\mathbf{x}_i, \mathbf{x}_{j'})$;

**4** $\quad \mathcal{N}_i \leftarrow \mathcal{E}_i \cup \{j\}$;

**5** $\quad \mathcal{C}_i \leftarrow \mathcal{C} \setminus \{j\}$;

**6** $\quad \mathcal{C}_i \leftarrow \{k \in \mathcal{C}_i \,|\, \text{connectivity rule between } i, j, \text{ and } k \text{ is met}\}$;

---

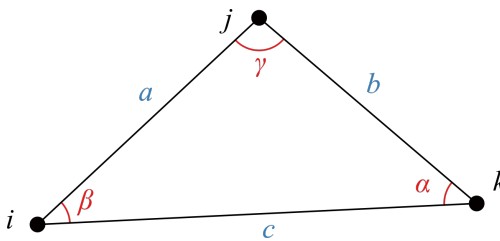

Figure 7: We show that, when using an RBF kernel, the graph pruning rules of most popular graph construction algorithms, inluding the popular HNSW (Malkov & Yashunin, 2020) and DiskANN (Subramanya et al., 2019), can be written as applications of the law of cosines and the inequality $a^2 + b^2 > c^2$.

**Theorem 5.** *Let $s, k$ be two distinct nodes of a graph built using Algorithm 4 with $\mathcal{C}_i = [1 \dots n] \setminus \{i\}$ and the connectivity rule in Lemma 3 and let $t = \underset{i \in V}{\operatorname{argmax}}\, K(\mathbf{x}_i, \mathbf{x}_k)$. There is a generalized 0-monotonic path between $s$ and $t$.*

The following corollary follows immediately from Definition 2 and Theorem 5.

**Corollary 3.** *A graph built using Algorithm 4 with $(\forall i)\, \mathcal{C}_i = [1 \dots n]$ and the connectivity rule in Proposition 1 is a generalized 0-monotonic search network.*

To the best of our knowledge, this is the first graph construction algorithm with full navigability guarantees regardless of the similarity function underlying the kernel. In this sense, it applies to any metric space and even non-metric vector spaces, e.g., spaces only equipped with an inner product.

For the RBF kernel, the connectivity rule in Lemma 3 amounts to the familiar inequality $a^2 + b^2 > c^2$, as depicted in Figure 7 and shown next.

**Corollary 4.** *When using the RBF kernel in the same setting as Proposition 1, we have*

$$K(\mathbf{x}_i, \mathbf{x}_j)K(\mathbf{x}_j, \mathbf{x}_k) < K(\mathbf{x}_i, \mathbf{x}_k) \quad \Leftrightarrow \quad \|\mathbf{x}_i - \mathbf{x}_j\|_2^2 + \|\mathbf{x}_j - \mathbf{x}_k\|_2^2 > \|\mathbf{x}_i - \mathbf{x}_k\|_2^2. \tag{11}$$

From Corollary 4, we can derive many graph construction rules by combining $a^2 + b^2 > c^2$ with different formulas from the law of cosines. Corollaries 5 and 6 provide two specific examples (in the following, we connect them with existing graph indices).

**Corollary 5** (Shekkizhar & Ortega, 2023)**.** *When using an RBF kernel in the same setting as Lemma 3, the necessary condition to connect node $i$ with node $k$ is*

$$(\cos \alpha) \|\mathbf{x}_i - \mathbf{x}_k\|_2 < \|\mathbf{x}_j - \mathbf{x}_k\|_2, \tag{12}$$

*where $\alpha$ is the angle between the vectors $\mathbf{x}_i - \mathbf{x}_k$ and $\mathbf{x}_j - \mathbf{x}_k$.*

**Corollary 6.** *When using an RBF kernel in the same setting as Lemma 3, the necessary condition to connect node i with node k is*

$$(\cos \beta) \left\| \mathbf{x}_i - \mathbf{x}_k \right\|_2 < \left\| \mathbf{x}_i - \mathbf{x}_j \right\|_2, \tag{13}$$

*where $\beta$ is the angle between the vectors $\mathbf{x}_k - \mathbf{x}_i$ and $\mathbf{x}_j - \mathbf{x}_i$.*

### 4.1 Graph indices under the lens

In the following, we show that the most popular graph indices can be interpreted as thresholded applications of different formulas from the law of cosines combined with $a^2 + b^2 > c^2$ (see Figure 7). In particular, our results cover the popular HNSW (Malkov & Yashunin, 2020) and DiskANN (Subramanya et al., 2019).

We start with the connectivity rule shared by MRNG (Fu et al., 2019) and HNSW (Malkov & Yashunin, 2020). For $i, j \in [1 \dots n]$, let

$$lune_{ij} = \left\{ \mathbf{x} \in \mathbb{R}^d \mid \left\| \mathbf{x}_i - \mathbf{x} \right\|_2 \leq \left\| \mathbf{x}_i - \mathbf{x}_j \right\|_2 \wedge \left\| \mathbf{x}_j - \mathbf{x} \right\|_2 \leq \left\| \mathbf{x}_i - \mathbf{x}_j \right\|_2 \right\} \tag{14}$$

An MRNG is a directed graph $G = (\mathcal{V}, \mathcal{E})$ with $\mathcal{V} = [1 \dots n]$ and the edge set $\mathcal{E}$ satisfying the following property: For any pair $i, k \in [1 \dots n]$, $\overrightarrow{ik} \in \mathcal{E}$ if and only if $lune_{ik} \cap S = \emptyset$ or $(\forall r) \, \mathbf{x}_r \in lune_{ik} \cap S \implies \overrightarrow{ir} \notin \mathcal{E}$. MRNG and HNSW use the following connectivity rule in Algorithm 4: Keep $k$ in $\mathcal{C}_i$ if

$$\left\| \mathbf{x}_i - \mathbf{x}_k \right\|_2 \leq \left\| \mathbf{x}_j - \mathbf{x}_k \right\|_2. \tag{15}$$

HNSW applies this rule within a hierarchical structure. A direct application of Corollary 5 yields the following result.

**Corollary 7.** *Running Algorithm 4 with the RBF kernel and the MRNG connectivity rule in Equation (15) is equivalent to applying the necessary condition in Corollary 5 with the additional simplification that $\cos \alpha = 1$.*

Vamana, the algorithm behind DiskANN (Subramanya et al., 2019), is an extension of MRNG (Fu et al., 2019) that seeks to accelerate the graph traversal speed. Recently, this acceleration has been formally proven in the worst case (Indyk & Xu, 2023). Vamana uses the following connectivity rule in Algorithm 4: keep $k$ in $\mathcal{C}_i$ if

$$\left\| \mathbf{x}_i - \mathbf{x}_k \right\|_2 \leq \lambda \left\| \mathbf{x}_j - \mathbf{x}_k \right\|_2. \tag{16}$$

A direct application of Corollary 5 yields the following result.

**Corollary 8.** *Running Algorithm 4 with the RBF kernel and the Vamana connectivity rule in Equation (16) is equivalent to applying the necessary condition in Corollary 5 with the additional simplification that $\alpha = \arccos \lambda^{-1}$.*

We also extend these results to the recently introduced SSG (Fu et al., 2022). For this, we use the following definitions

$$\text{angle}(\mathbf{x}, \mathbf{y}) = \arccos \frac{\langle \mathbf{x}, \mathbf{y} \rangle}{\left\| \mathbf{x} \right\|_2 \left\| \mathbf{y} \right\|_2}, \tag{17}$$

$$\text{ball}(i, \delta) = \left\{ \mathbf{x} \in \mathbb{R}^d \mid \left\| \mathbf{x} - \mathbf{x}_i \right\|_2 \leq \delta \right\}, \tag{18}$$

$$\text{cone}_{ij}^{\theta} = \left\{ \mathbf{x} \in \mathbb{R}^d \mid \text{angle}(\mathbf{x} - \mathbf{x}_i, \mathbf{x}_j - \mathbf{x}_i) \leq \theta \right\}. \tag{19}$$

An SSG is a directed graph $G = (\mathcal{V}, \mathcal{E})$ with $\mathcal{V} = [1 \dots n]$ and the edge set $\mathcal{E}$ satisfying the following property: For any pair $i, k \in [1 \dots n]$, $\overrightarrow{ik} \in \mathcal{E}$ if and only if $\text{cone}_{ik}^{\theta} \cap \text{ball}(i, \left\| \mathbf{x}_i - \mathbf{x}_k \right\|_2) \cap S = \emptyset$ or $(\forall r, \mathbf{x}_r \in \text{cone}_{ik}^{\theta} \cap \text{ball}(i, \left\| \mathbf{x}_i - \mathbf{x}_k \right\|_2) \cap S) \, \overrightarrow{ir} \notin \mathcal{E}$, where $0 \leq \theta \leq 60°$ is a hyperparameter. SSG uses the following connectivity rule in Algorithm 4: for $0 \leq \theta \leq 60°$, keep $k$ in $\mathcal{C}_i$ if

$$\text{angle}(\mathbf{x}_j - \mathbf{x}_i, \mathbf{x}_k - \mathbf{x}_i) \geq \theta \vee \left\| \mathbf{x}_j - \mathbf{x}_i \right\|_2 \geq \left\| \mathbf{x}_i - \mathbf{x}_k \right\|_2. \tag{20}$$

A direct application of Corollary 6 yields the following result.

---

**Algorithm 5:** Subspace pursuit for SVG construction

---

    **Input**   : Dataset $\mathcal{X} = \left\{ \mathbf{x}_j \in \mathbb{R}^d \right\}_{j=1}^n$, element $i \in [1 \dots n]$, maximum out-degree $M \in \mathbb{N}^+$.

    **Output:** Set $\mathcal{N}_i$ of outgoing neighbors for node $i$.

**1** Select the kernel width $\sigma$ used for node $i$;

**2** $\mathcal{N}^{(0)} \leftarrow \emptyset$;

**3 for** $t \in [1 \dots T]$ **do**

**4**     Let $\mathcal{C}$ be the set of the $M$ largest entries in

$$\left\{ K(\mathbf{x}_i, \mathbf{x}_k) - \sum_{j \in \mathcal{N}^{(t-1)}} s_j^{(t-1)} K(\mathbf{x}_j, \mathbf{x}_k) \,\middle|\, k \in [1 \dots n] \right\} \tag{21}$$

**5**     $\mathcal{C} \leftarrow \mathcal{C} \cup \mathcal{N}^{(t-1)}$;

**6**     Find the solution $\mathbf{s}$ to Problem (22), restricted to vectors in $\mathcal{C}$;

**7**     Let $\mathcal{N}^{(t)}$ be the set of indices corresponding to the M largest entries of $\mathbf{s}$;

**8**     $\mathcal{N}^{(t)} \leftarrow \left\{ j \,\middle|\, s_j^{(t)} > 0 \right\}$;

**9**     **if** $\mathcal{N}^{(t)} = \mathcal{N}^{(t-1)}$ **then** break;

**10** $\mathcal{N}_i \leftarrow \mathcal{N}^{(t)}$;

---

**Corollary 9.** *Running Algorithm 4 with the RBF kernel and the SSG connectivity rule in Equation (20) rule is equivalent to applying the necessary condition in Corollary 6 with the additional simplification that* $\theta = \arccos \left( \left\| \mathbf{x}_i - \mathbf{x}_j \right\|_2 / \left\| \mathbf{x}_i - \mathbf{x}_k \right\|_2 \right)$.

In summary, the edge pruning rules used in Algorithm 4 by some of the most popular graph indices can be regarded as specializations of the SVG optimization. These specializations, described in Figure 7, emerge from applying the optimization to triplets of points.

## 5   Fast SVG construction with bounded out-degree

Graph construction algorithms based on Algorithm 4, such as those described in Section 4, have two main practical issues that require careful tuning.

First, Algorithm 4 requires a candidate pool $\mathcal{C}_i$. Setting $\mathcal{C}_i = [1 \dots n] \setminus \{i\}$ provides theoretical guarantees, as described in Section 4, but becomes untenable as $n$ grows. In practice, $\mathcal{C}_i$ is heuristically determined by finding the (approximate) nearest neighbors of $\mathbf{x}_i$. However, this may be problematic if, for example, $\mathbf{x}_i$ lies on the outskirts of a tight cluster (Indyk & Xu, 2023) as the graph may become disconnected. Take the example of the red point in Figure 9. We would need to create a candidate pool larger than the number of points in the left cluster for Algorithm 4 to ensure navigability between both clusters.[2] Finding a prudent size for $\mathcal{C}_i$ becomes a dataset-specific tuning problem.

Second, although Algorithm 4 produces sparse graphs when paired with a suitable edge selection rule (such as those in Section 4), the graphs are generally not sparse enough. The sparsity of these graphs is critical as it directly determines the footprint and the search runtime of the index. Let $M$ be the maximum out-degree we want in a graph. Because Algorithm 4 neither produces a total order nor handles the cardinality constraint intrinsically, the list of neighbors is truncated in an ad-hoc fashion by stopping Algorithm 4 once $|\mathcal{N}_i| = M$. This heuristic may cause navigability problems, as described in Figure 8. In essence, the diversity of the selected edges becomes suboptimal, and possibly non-navigable, if the process is terminated early.

In this section, we show that the SVG framework overcomes these difficulties. Although the solution to Problem (4) is naturally sparse, we would like to impose a more stringent and specific level of sparsity to

---

[2]The graph construction by Shekkizhar & Ortega (2023) that uses Problem (4) for manifold learning shares these issues.

bound the out-degree of the graph. Moreover, we show that once this restriction is added, precomputing a candidate pool becomes unnecessary.

We address both problems simultaneously by altering the SVG optimization. We bound the sparsity level by solving the related problem

$$\min_{\mathbf{s}} \frac{1}{2} \left\| \phi(\mathbf{x}_i) - \mathbf{\Phi s} \right\|_2^2 \quad \text{s.t.} \quad \mathbf{s} \geq \mathbf{0}, \ \|\mathbf{s}\|_0 \leq M, \tag{22}$$

where the so-called $\ell_0$ norm measures the number of non-zero entries of a vector. We use Problem (22) to build a graph with a maximum out-degree, since the minimizer $\mathbf{s}_i^*$ will have at most $M$ nonzero entries. In contrast with the typical truncation heuristic, Problem (22) selects the subset of $M$ elements that provides the best tradeoff between diversity and similarity (see the analysis of Problem (5)). This approach has connections with sparse SVMs (Smola et al., 1999), which use parsimony-inducing $\ell_1$ (e.g., Bi et al., 2003) or $\ell_0$ (e.g., Zhang et al., 2023) constraints to sparsify the support vector set.

The astute reader will notice that solving problems (4) and (22) quickly becomes impractical as $n$ grows: The size of the kernel matrix $\mathbf{K}$ is quadratic in $n$ and we solve an optimization with $n$ variables. It seems like our computational requirements are still very high. Additionally, problems involving $\ell_0$ constraints have always been considered challenging because of their non-convexity and NP-hardness. The dominant paradigm replaces these constraints by convex $\ell_1$ constraints (e.g., Candès & Plan, 2009).[3] However, Problem (22) belongs to a particular family of problems, known as subspace pursuit, for which there are very efficient algorithms (Dai & Milenkovic, 2009; Needell & Tropp, 2010). Algorithm 5 presents an algorithm that solves this problem. Additional details on subspace pursuit can be found in Section B.

Algorithm 4 does not depend on precomputing an appropriate candidate pool, in contrast to Algorithm 2. This superpower comes from Line 4, which performs a neighbor search with a modified similarity. By finding the vectors that maximize

$$K(\mathbf{x}_i, \mathbf{x}_k) - \sum_{j \in \mathcal{N}^{(t-1)}} s_j^{(t-1)} K(\mathbf{x}_j, \mathbf{x}_k), \tag{23}$$

it becomes clear that the similarity in this neighbor search selects vectors that are close to $\mathbf{x}_i$ and far away from the vectors in $\mathcal{N}^{(t-1)}$. This search focuses its attention on portions of the space not considered in previous iterations (see Figure 9).

As a side note, by writing Equation (23) as

$$\mathbf{v}^\top \phi(\mathbf{x}_k), \quad \text{where} \quad \mathbf{v} = \phi(\mathbf{x}_i) - \sum_{j \in \mathcal{N}^{(t-1)}} s_j^{(t-1)} \phi(\mathbf{x}_j), \tag{24}$$

the computation can be carried out using random features (RF) (Rahimi & Recht, 2007; Reid et al., 2023; Liu et al., 2022; Sernau et al., 2024). However, the literature has understandably concentrated on approximating the central portion of kernels instead of their tails (e.g., for exponential kernels, where $K_{\text{EXP}}(\mathbf{x}, \mathbf{x}') \approx 0_+$). Since Equation (24) is concerned with the tails (notice the small values in the attention area in Figure 9), new RF techniques would be needed. This is an interesting future line of work.

**Computational complexity.** Algorithm 5 involves solving a least squares problems on the simplex in Line 6 with $2M$ variables. Since this problem is convex and $M \ll n$, the computational complexity is drastically reduced from $O(n^3)$ to $O(M^3)$. Implementing Line 4 in Algorithm 5 using a vector search index (e.g., by searching the SVG-L0 graph as we incrementally build it), the complexity of Algorithm 5 becomes sublinear in the number of indexed vectors. All in all, SVG-L0 has complexity similar to other existing graph indices (Malkov & Yashunin, 2020; Fu et al., 2019; Subramanya et al., 2019).

---

[3]Direct $\ell_0$ solvers have gained significant attention in the last few years (Bertsimas et al., 2016; Hastie et al., 2020) due to improvements in mixed integer optimization.

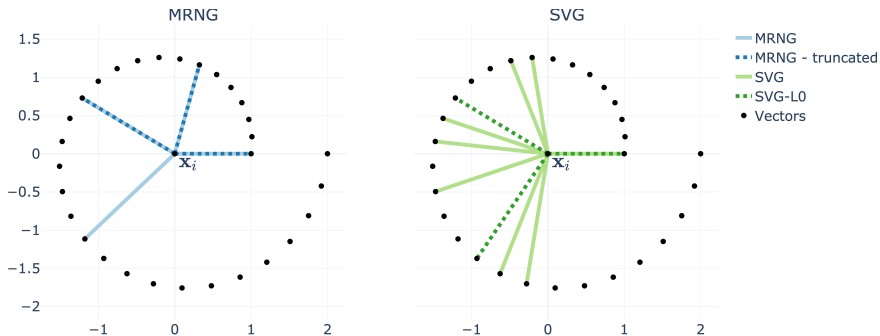

Figure 8: Although the MRNG is sparse, it is not sparse enough in practical situations and its list of neighbors is truncated by stopping Algorithm 4 early after $\mathcal{N}_i$ attains a prescribed size. With the resulting edge set (represented by dotted blue lines on the left plot), navigating downwards from $\mathbf{x}_i$ is not possible. In contrast, by using Problem (22) to build the degree-constrained SVG, we obtain an edge set (represented by dotted green lines on the right plot) with improved diversity and thus navigability. Note that the SVG-L0 edges are not necessarily a subset of the SVG edges.

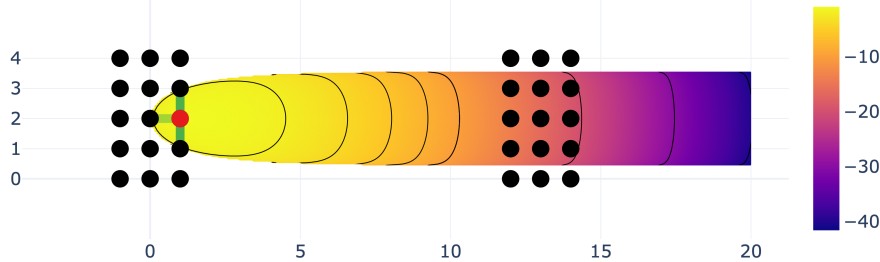

Figure 9: The attention of the search driven by Equation (23) after connecting the red point to its three neighbors (green edges). We show the values of Equation (23) in a logarithmic colorscale in the area where it is positive. The attention will focus on finding the next $M$ points to the right of the red point, seeking to find another support vector for the SVM classifier.

## 6    Experimental results

We present a few experimental results to highlight the practical value of SVG and SVG-L0. We use the standard recall@1 measure of search accuracy, which counts how many times we find the ground truth NN of every vector $\mathbf{x}_i \in \mathcal{X}$ when using it as the query.

In addition to purely greedy graph search algorithms, we also experimented with backtracking since it is widely used in practice. Here, we are interested in observing how quasi-navigable networks behave when combined with this common "trick." We use a small queue with a length of 2.

Our results in Section 3.1 predict that a small kernel width is needed to guarantee navigability when using an exponential. This is empirically verified by the experiment in Figure 10, where the empirical navigability (recall) increases as $\sigma$ decreases. In this small-$\sigma$ setting, the SVG becomes very navigable (recall close to 1). The use of backtracking allows us to extend similar accuracies with suboptimal selections of $\sigma$.

We also compared SVG-L0 (Section 5) with the degree-constrained MRNG and the truncated MRNG. As discussed in Section 4, the MRNG is a fully navigable network with no degree constraints. This feature comes at a steep price: the complexity of building with MRNG is $O(n^3)$. Here, we use a (still computationally costly) variant that uses a maximum out-degree $M$ and an unlimited candidate pool size, resulting in a complexity of $O(n^2)$. The truncated MRNG has the additional constraint of working with a fixed candidate pool size (see Algorithm 2). Note that the truncated MRNG is the algorithm that is used in practice today to build graph indices. We set the size of the candidate pool as a multiplicative factor of $M$, that is, $|\mathcal{C}| = rM$ for $r > 1$. SVG-L0 uses a maximum out-degree but does not need a candidate pool.

As shown in Figure 11, SVG-L0 is competitive with the degree-constrained MRNG when working with randomly distributed vectors. Its accuracy is slightly lower in the greedy setting, but slightly higher in the backtracking setting. Both indices are clearly superior to the truncated MRNG.

Lastly, we experimented with a few small real-world datasets using Python implementations of SVG and SVG-L0 that were not optimized to scale. We took $10^4$ vectors from the datasets colbert-1M, cohere-english-v3-100k, and openai-v3-small-100k with $d = 128, 1024, 1536$ dimensions, respectively (`https://github.com/datastax/jvector`). We show in Figure 12 that the navigability slack $\epsilon$ in SVG, given in Theorem 3, is relatively small and that Algorithm 3 often succeeds. In addition, we built indices with SVG-L0 and with the truncated MRNG described previously. We observe in Figure 13 that SVG-L0 outperforms the truncated MRNG, even when using a large candidate size ($|\mathcal{C}| = rM$ with $r = 8$). The effect is more pronounced in higher dimensions, where the value of $\sigma$ in SVG-L0 has little effect on its accuracy. Interestingly and unsurprisingly according to our theoretical results, the accuracy of SVG-L0 has a link with the distribution of $\epsilon_i$ (the smaller the mean in Theorem 3, the better the accuracy in Figure 13).

## 7 Conclusions and future work

We introduced a new type of graph index, the Support Vector Graph (SVG). We derive SVG from a novel perspective that uses machine learning instead of computational geometry to build the index. Concretely, we have formulated the graph construction as a kernelized nonnegative least squares problem. This problem is in turn equivalent to a support vector machine, whose support vectors yield the connectivity of the graph.

We extended the notion of graph navigability to allow for approximate navigation, which is more in line with the practical use of graphs as ANN indices. We provide formal quasi-navigability and, under some conditions, navigability results for SVG that are valid in any metric and non-metric vector spaces (e.g., for inner product similarity). To the best of our knowledge, these are the first navigability results in non-Euclidean spaces.

We formally interpreted the most popular graph indices, including HNSW (Malkov & Yashunin, 2020) and DiskANN (Subramanya et al., 2019), as SVG specilizations. We also showed that new traditional (i.e., triangle-pruning) algorithms can be derived from the principles behind this specialization.

Finally, we showed that we can build graphs with a bounded out-degree by adding a sparsity ($\ell_0$) constraint to the SVG optimization, a combination that we name SVG-L0. SVG-L0 yields a principled way of handling the bound, in contrast to the traditional heuristic of simply truncating the out edges of each node. Additionally, SVG-L0 has a self-tuning property, which avoids selecting a candidate set of edges for each graph node and makes its computational complexity sublinear in the number of indexed vectors.

In future work, we plan to address the following issues to further improve SVG and SVG-L0. First, tuning the kernel width, although not hard and standard in SVMs (e.g., Chapelle et al., 2002), can be challenging in large-scale scenarios. We will address this problem, seeking an automated selection that does not require cross-validation. Second, we will further accelerate the optimization of Problem (4) with more specific algorithms (our current implementation uses an off-the-shelf solver). Additionally, and as aforementioned, we posit that the formal study of quasi-navigable graphs is perhaps of greater practical relevance for ANN than that of exactly navigable graphs. This is an exciting future line of work. Lastly, it remains to be seen whether other machine learning techniques, beyond kernel methods, can be utilized to build graph indices.

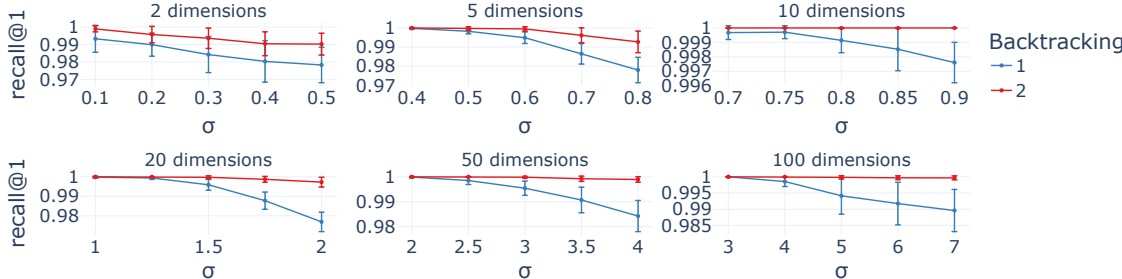

Figure 10: As Corollary 1 predicts, SVG's empirical navigability improves as $\sigma$ decreases. We compute the recall@1 over ten realizations of 100 random vectors with different numbers of dimensions. As typical in practical deployments, we include the results of using a backtracking queue (of length 2). In low dimensions, the SVG navigability is more impacted by the width; in higher dimensions this effect is significantly milder (probably due to the growing average out-degree, see Figure 5).

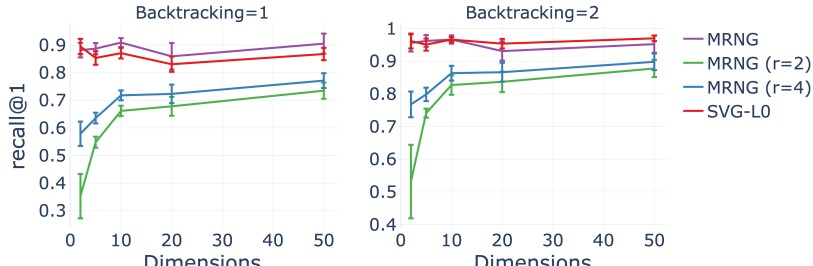

Figure 11: The low-complexity SVG-L0, defined in Problem (22), offers better empirical navigability than that of the truncated MRNG (with similar complexity) and competitive with the degree-constrained MRNG (with a quadratic complexity in the number of vectors). We compute the recall@1 over ten realizations of 100 random vectors with different numbers of dimensions. As typical in practical deployments, we include the results of using a backtracking queue (of length 2). For each dimension, we selected the maximum out-degree $M$ that yields reasonable performance (around 85%) for the MRNG. For the truncated MRNG, we define the truncation ratio $r = |\mathcal{C}|/M$, where $\mathcal{C}$ is the candidate pool.

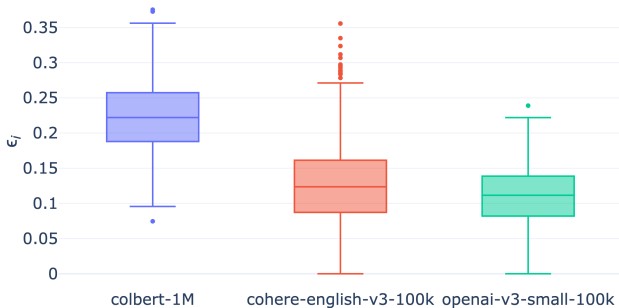

Figure 12: Theorem 3 predicts that SVG is a generalized $\epsilon$-navigable network with $\epsilon = \max_{i \in [1..n]} \epsilon_i$. Using the RBF kernel with width $\sigma = 1$, the distribution of the values $\epsilon_i$ is relatively tight and has small mean for $10^3$ vectors from different datasets. In practice, Algorithm 3 often succeeds and any errors are fixed with a small amount of backtracking.

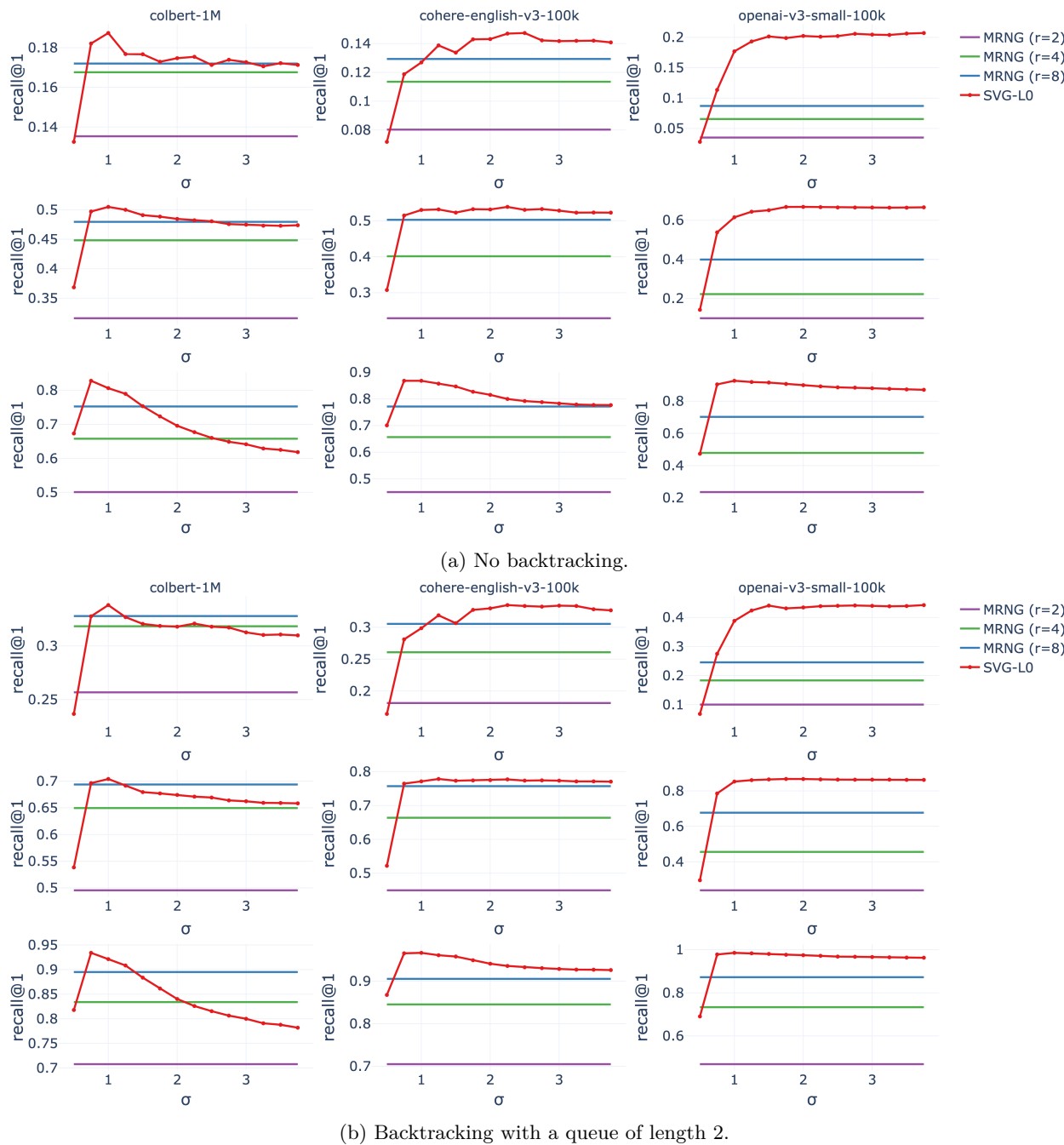

Figure 13: The SVG-L0, defined in Problem (22), offers better empirical navigability than that of the truncated MRNG. We compute the recall@1 for different datasets (columns) and maximum out-degrees $M = 8, 16, 32$ (top, center, and bottom rows, respectively). For the truncated MRNG, we define the truncation ratio $r = |\mathcal{C}|/M$, where $\mathcal{C}$ is the candidate pool. In practice, finding a suitable $\sigma$ for SVG-L0 is not difficult (automating the selection is left for future work).

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
