# OpenReview forum: "The kernel of graph indices for vector search"
_TMLR — Rejected by TMLR_

### Review · Reviewer_U2up · 2025-07-31

**Summary Of Contributions:**

The paper proposes the use of Support Vector Graph (SVG) for the construction of graph indices used in the context of vector search considering both metric but importantly also non-metric vector spaces notably providing formal navigability guarantees. The key insight is to replace the greedy search with a multiclass SVM type optimization problem forming the SVG. The SVG is in turn defined by the cone (i.e. non-negative least squares regression) that best spans the feature vector of each node based on the feature vectors of the other observations, which due to the least-squares objective naturally can be written in terms of the associated Kernel. Notably for non-negative kernels (such as the RBF) the angle is less than pi/2 and this optimization can in turn be related to SVM and thus interpreted in its solution as support vectors with relevant associated algorithms. The approach nicely introduce the relevant theoretical properties of the developed framework and include some limited experimentation demonstrating the utility of the SVG and SVG-L0 (i.e., further pruned) approach.

**Audience:**

Yes

**Broader Impact Concerns:**

There are no direct broader impact concerns of this theoretical paper proposing a new methodology for vector search.

**Claims And Evidence:**

Yes

**Requested Changes:**

I think the paper is generally well written and clear and the contribution seems solid and the approach novel in terms of the methodology used for vector search. I think the paper can be somewhat strengthened in particular by improving the experimentation.

Points for clarification and potential improvements:

 “For a graph with n nodes, the number of edges in the DG rapidly approaches n as the dimensionality grows” shouldn’t this be “number of edges per node” instead of “number of edges” which would also result in memory use as stated by O(n^2).

In section 2 what is mathcal{C}_i used to denote? This is not introduced as notation. I assume it defines the candidate pool as specified in Algorithm 2.

In the statement: “(i.e., preventing the nonnegative least squares from overfitting).” It is unclear how uniqueness relates to overfitting, i.e. that a minima is unique may in general not ensure such minima is not an overfitted solution. This needs to be clarified. Furthermore, the sparsity interpreted as regularization I believe depends on the associated dimensionality of the associated Hilbert space defining the kernel despite all vectors residing (through rotation) in the positive orthant.

The paper states: “Algorithm 5 involves solving a least squares problems on the simplex in Line 6 with 2M variables.”
It is unclear to me why this problem is on the simplex this needs to be clarified – the standard simplex would imply a constraint on the l1 norm which I do not see here have been invoked.

Furthermore, would full \ell_1 regularization paths solutions such as described in:

Mørup, Morten, Kristoffer Hougaard Madsen, and Lars Kai Hansen. "Approximate l 0 constrained non-negative matrix and tensor factorization." 2008 IEEE International Symposium on Circuits and Systems (ISCAS). IEEE, 2008.

using the least angle regression and selection (LARS) framework of

B. Efron, T. Hastie, I. Johnstone and R. Tibshirani Least angle regression. Annals of Statistics, 32(2):407–499, 2004.

to define the entire regularization path under non-negativity be useful as proxy here to find the minimal \ell_0 solution as opposed to subspace pursuit?

Can the approach be benchmarked and positioned wrt. the many existing graph vector search based procedures as described and evaluated for instance in:

https://arxiv.org/html/2502.05575v1

**Strengths And Weaknesses:**

Strengths:

•	The approach is interesting and well described and although the use of non-negative least squares based graph construction has been considered previously as outlined in the manuscript the present connection to vector based search is new and interesting.

•	The SVG procedure and SVG-L0 procedure appears promising and attractive.

•	Theoretical insights and properties are well described and is a strongpoint of the contribution.

•	The paper is generally well written and the approach appears novel in the context of vector search.


Weaknesses

•	A few points are unclear and needs clarification (see requested changes).

•	The experimentation is rather limited and the performance of the vector search not systematically contrasted SOTA methods. Arguably the scope is also more to introduce theoretical properties but this would strengthen and position the approach better (see  requested changes).

---

### Review · Reviewer_TcZ1 · 2025-08-13

**Summary Of Contributions:**

Graph-based indexes for nearest neighbor search are widely popular, making up the foundation of popular libraries like FAISS, DISK-ANN, etc. These structures operate by navigating through the graph (using greedy/beam search) with a given query point, exploring neighboring points in the graph to "descend" the structure to find the points nearest to the query. These structures, while widely used, are hardly understood. This paper address this want of understanding by analyzing graph indices using a classification-based framing. This framing results in the Support-Vector-Graph (SVG), for which the authors provide detailed description, analysis, and supporting experiments. SVG draws connections between many clearly related topics (clustering, dictionary learning, graph indices) and writes down and expands upon many things at this intersection that I have heard discussed among researchers (but seldom written down).

**Audience:**

Yes

**Claims And Evidence:**

Yes

**Requested Changes:**

* Discussion of related methods w/ data assumptions, eg Cover Tree, Navigating Nets, etc. Maybe also worth mentioning nearest neighbor descent since this had been popular also.
* If possible, some measurement of graph quality compared to popular heuristic methods.

**Strengths And Weaknesses:**

Strengths:
* As indicated in the above summary of contributions, the core strength of this paper is the connections it draws between related topics in clustering, vector coding, graph indices. This is quite well done by the authors. I found the paper engaging, understandable, and informative.
* The formulation of SVG is quite interesting and can inspire future work in understanding how such graph indices can be constructed. I think the importance of the way SVG connects different parts of machine learning should not be under-estimated. I think these connections are done well.
* In all, this seems like exactly the kind of paper I would expect to find in TMLR. It is well written, thorough, and informative.

Weaknesses:
* I would have liked to see the authors more explicitly describe time complexity. I think this is quite important for nearest neighbor search. It would be interesting to understand both in bounded & non-degree bounded cases.
* I think it is worth the paper discussing results regarding [Cover Trees](https://homes.cs.washington.edu/~sham/papers/ml/cover_tree.pdf) and [Navigating Nets](https://homes.cs.washington.edu/~jrl/papers/pdf/nn-soda.pdf). These papers make assumptions about the data to give time and space bounded algorithms for nearest neighbor search. Granted, these assumptions are more limited in several ways than the results of this paper, but are nevertheless I think important to give the reader a more complete sense of what is known.
* It would be nice to see the gap in performance between heuristic graph construction methods like hnsw / pynndescent / faiss etc and the graphs constructed by these methods -- e.g. measure number of distance computations required in search vs recall.

---

### Review · Reviewer_NuqR · 2025-08-18

**Summary Of Contributions:**

The paper introduces the Support Vector Graph (SVG) as a new construction for graph-based vector search. SVG is derived from a kernelized nonnegative least squares (NNLS) problem. Graph connectivity is defined by the support vectors of this optimization.

The main theoretical contribution is a set of navigability guarantees beyond Euclidean settings:

* For general PSD kernels, SVG graphs are shown to be **quasi-navigable**, i.e., there exists a path that is $\rho$-monotonic with explicit $\rho$ derived from the optimization coefficients.
* For the **exponential kernel**, in the limit of small width $\vartheta$, the slack $\rho \to 0$, yielding full navigability by greedy search. This according to the authors provides the first formal navigability results in non-Euclidean vector spaces, e.g., inner product similarity.

The authors further show that popular indices such as HNSW can be interpreted as specializations of SVG when its connectivity rule is restricted to triplets.

Finally, the paper introduces **SVG-L0**, which adds an explicit $\ell_0$ sparsity constraint to bound node out-degree. The claimed novelty here is a principled alternative to heuristic truncation, with a “self-tuning” property that avoids candidate pool heuristics and is argued to yield **sublinear construction complexity**. Empirically, SVG-L0 outperforms truncated MRNG baselines and matches degree-constrained MRNG while being cheaper to build.

**Audience:**

Yes

**Claims And Evidence:**

No

**Requested Changes:**

## Critical (blocking)

1. **Make the $\rho$ certificate non‑vacuous.**

   * Can you state explicitly in the main text the two formulas you actually use to certify progress, and why they are meaningful?

     * General PSD kernel: $\rho_i=\max\{\mathbf 1^\top s^{(i)},1\}-1$
     * Exponential kernels: $\rho_i\le |I_i|\cdot \max_{j\in I_i}s^{(i)}_j-1$
   * For **RBF** with $K(x,x)=1$, the KKT stationarity immediately gives $s_t\le 1$ for any active neighbor $t$ (supp. Eq. (46)), hence the simple a‑priori bound $\rho_i\le |I_i|-1$. Can you add this explicit bound in §3.1 (not just in the supplement)?



2. **Spell out the kernel assumptions wherever a guarantee is claimed.**

   * Which results **require PSD** ? Please mark those statements explicitly. &#x20;
   * Section 4 (Algorithm 4 + Lemma 3 + Thm. 5) **uses normalized kernels $K(x,x)=1$** to get strict improvement. Please state this **assumption** at the start of §4.
   * What, concretely, breaks for **indefinite** similarities (no RKHS)? A short remark (e.g., convexity loss, failure of the geometry behind $f_i$) would be useful.

3. **Rephrase the ANN analogy.**

   * Right now $\rho$-monotonicity is compared to $(1+\varepsilon)$-ANN. But $(1+\varepsilon)$ is a **global** approximation on the **returned neighbor**, while $\rho$ is a **local step‑quality** bound. Can you **clarify this** and avoid implying equivalence of guarantees?

---

## Important (not blocking)

4. **SVG‑L0 complexity: what exactly is “sublinear”?**

   * In Algorithm 5, Line 6 solves a convex NNLS in **$2M$** variables (so $O(M^3)$ per iteration). The **sublinear** claim seems to come from Line 4’s neighbor search done over an index (Eq. (23)).  Please **define the model** (index used, assumptions on recall) and scope the claim precisely.

5. **Lemma 3 phrasing and provenance.**

   * Lemma 3 is not formally written, what is the meaning of "connectivity rule derived from Problem (4)"?

6. **Delaunay graph wording.**

   * p. 3 says “the number of edges in the DG rapidly approaches $n$ as dimensionality grows,” which reads like total edges $\sim n$. Please **clarify** (average degree vs total edges).

7. **Connectivity: state where it actually comes from.**

  * Can you give a simple argument (without appealing to approximate navigability guarantees) as to why the graph resulting from SVG is connected?


8. **What do we buy over the complete graph?**

* A complete graph is trivially 0‑MSN but has quadratic number of edges. Can you add two sentences clarifying what SVG buys: **sparsity** with an **explicit $\rho$ certificate**. Can you provide a-priori bound (eg. O(n)) on the sparsity non-degenerate cases?

9. **Pruning guarantees vs thresholded heuristics.**

* In §4 you prove 0‑MSN for the **kernel rule** (Lemma 3/Thm. 5), then recover HNSW/MRNG and Vamana as **thresholded** law‑of‑cosines conditions (Cor. 7–8). Can you make the **gap** explicit: i.e., the thresholded rules are necessary‑condition simplifications and **do not** inherit the full 0‑MSN guarantee unless the full kernel rule is used with full candidates?

**Strengths And Weaknesses:**

## Strengths

* **Kernel/SVM lens with explicit certificates.** The per‑node **computable** $\rho_i$ (Theorems 3–4) is not just “there exists a $\rho$”: it is read off from $s^{(i)}$.

* **Non‑Euclidean guarantees.** Quasi‑navigability for **general PSD** kernels and **0‑monotone** pruning (Algorithm 4) for **normalized** kernels—both extend beyond Euclidean geometry

* **Unified view of pruning.** The triplet rule + law‑of‑cosines derivations explain popular algorithms like HNSW as **thresholded** instances (Cor. 7–8).

---

## Weaknesses / points that confused me

* **“$\rho$-monotonicity is trivial” worry.** As written, it’s easy to miss why this is not vacuous. The paper should say why the given bound on $\rho$ is meaningful. It is stated in terms of $s_i$ and it is not clear if kernel NNLS algorithm for constructing the graph leads to a better $\rho$ than some other popular methods.

* **ANN analogy.** $(1+\varepsilon)$-ANN is a **global** approximation guarantee on the returned neighbor; $\rho$-monotonicity is a **local step‑quality bound**. I’d avoid suggesting they are analogous guarantees; they are different objects.

* **$\rho$-monotonicity and greedy.** There is no formal guarantees for the greedy algorithm (Algorithm 3) on $rho$-monotonic graphs.

* **SVG‑L0 “sublinear” claim.** The text says “sublinear in the number of indexed vectors.” What is sublinear? The **per‑iteration neighbor search** in Line 4 of Algorithm 5 (when run over an index) can be sublinear; total build is still at least $n$. Please pin down the model of computation and whether “sublinear” is per node, per iteration, or overall.

* **Lemma 3 phrasing.** It currently reads like a definition. The supplement actually **derives** the rule from Problem (4) on a triplet (Proposition 1 → Lemma 3). Say this and cite the equations (66)–(80). Also, call the Euclidean form “Pythagoras/law‑of‑cosines,” not “triangle inequality.”

* **Delaunay discussion.** “Number of edges in the DG rapidly approaches $n$” (p. 3) is ambiguous. Do you mean **average degree** grows toward $n$ (graph becomes dense), hence **total edges** approach $O(n^2)$?

---

### Decision · Action_Editor_Jojo · 2025-10-09

**Recommendation:** Reject

**Additional Comments:**

Unfortunately, I must recommend to reject the paper.  Some reviewers provided (what seems like fairly minor) but critical requests.  And the authors did not reply or address them.  If they do, then I think the paper can and should probably be accepted.

**Audience:**

Yes

**Audience Explanation:**

The topic of understanding when and why graph-based NN search can work is a really exciting and important area, very relevant to TMLR.

**Claims And Evidence:**

No

**Claims Explanation:**

I am torn on this.  The paper is generally in pretty good shape, and is for the most part liked by the reviewers.  However, each reviewer asked for some fairly minor seeming changes.  Some of them provide some fairly small but important points on accuracy and clarify.  The authors unfortunately, did not reply or address any of them.  So I have settled on saying "No" for question.

**Resubmission Of Major Revision:**

The authors may consider submitting a major revision at a later time.

---

> ### Author Response · Authors · 2025-10-09
> **Follow-up**
>
> We thank the reviewers and the AE for their positive comments and useful feedback. We agree that the contributions in the paper have value and think that addressing the reviewer's comments would make it stronger. We also would like to apologize for the delay in submitting the revised manuscript. We were working through the changes when we received this update and didn't realize there was a specific deadline. We will certainly complete the revision in short order and resubmit as suggested.